# Mirror Descent Maximizes Generalized Margin and Can Be Implemented Efficiently

**Haoyuan Sun**
MIT
haoyuans@mit.edu

**Kwangjun Ahn**
MIT
kjahn@mit.edu

**Christos Thrampoulidis**
UBC
cthrampo@ece.ubc.ca

**Navid Azizan**
MIT
azizan@mit.edu

## Abstract

Driven by the empirical success and wide use of deep neural networks, understanding the generalization performance of overparameterized models has become an increasingly popular question. To this end, there has been substantial effort to characterize the implicit bias of the optimization algorithms used, such as gradient descent (GD), and the structural properties of their preferred solutions. This paper answers an open question in this literature: For the classification setting, what solution does mirror descent (MD) converge to? Specifically, motivated by its efficient implementation, we consider the family of mirror descent algorithms with potential function chosen as the $p$-th power of the $\ell_p$-norm, which is an important generalization of GD. We call this algorithm $p$-GD. For this family, we characterize the solutions it obtains and show that it converges in direction to a *generalized maximum-margin* solution with respect to the $\ell_p$-norm for linearly separable classification. While the MD update rule is in general expensive to compute and perhaps not suitable for deep learning, $p$-GD is fully parallelizable in the same manner as SGD and can be used to train deep neural networks with virtually *no additional computational overhead*. Using comprehensive experiments with both linear and deep neural network models, we demonstrate that $p$-GD can noticeably affect the structure and the generalization performance of the learned models.

## 1 Introduction

Overparameterized deep neural networks have enjoyed a tremendous amount of success in a wide range of machine learning applications [Brown et al., 2020, Dosovitskiy et al., 2020, Ramesh et al., 2021, Schrittwieser et al., 2020]. However, as these highly expressive models have the capacity to have multiple solutions that interpolate training data, and not all these solutions perform well on test data, it is important to characterize which of these interpolating solutions the optimization algorithms converge to. Such characterization is important as it helps understand the generalization performance of these models, which is one of the most fundamental questions in machine learning.

Notably, it has been observed that even in the absence of any explicit regularization, the interpolating solutions obtained by the standard gradient-based optimization algorithms, such as (stochastic) gradient descent, tend to generalize well. Recent research has highlighted that such algorithms favor particular types of solutions, i.e., they *implicitly regularize* the learned models. Importantly, such implicit biases are shown to play a crucial role in determining generalization performance, e.g., [Donhauser et al., 2022, Neyshabur et al., 2014, Wilson et al., 2017, Zhang et al., 2021].

In the literature, the implicit bias of first-order methods is first studied in linear settings since the analysis is more tractable, and also, there have been several theoretical and empirical evidence that certain insights from linear models translate to deep learning, e.g. [Allen-Zhu et al., 2019, Bartlett et al., 2017, Belkin et al., 2019, Jacot et al., 2018, Lyu and Li, 2019, Nakkiran et al., 2021]. In the linear setting, it is easier to establish implicit bias for regression tasks, where square loss is typically

36th Conference on Neural Information Processing Systems (NeurIPS 2022).

Table 1: **Conceptual summary of our results.** In the case of linear regression, the implicit regularization results are complete; it is shown that mirror descent converges to the interpolating solution that is closest to the initialization. However, such characterization in the classification setting is missing in the literature and this is precisely the goal of this work. In particular, motivated by its practical application, we consider the potential function $\psi(\cdot) = \frac{1}{p} \left\| \cdot \right\|_p^p$ and extend the result of the gradient descent to such mirror descents.

|  | Regression | Classification |
|---|---|---|
| Gradient Descent $(\psi(\cdot) = \frac{1}{2} \left\| \cdot \right\|_2^2)$ | $\mathrm{argmin}_w \left\| w - w_0 \right\|_2$ 
 s.t. $w$ fits all data 

 [Engl et al., 1996, Thm 6.1] | $\mathrm{argmin}_w \left\| w \right\|_2$ 
 s.t. $w$ classifies all data 
 Soudry et al. [2018] 
 Ji and Telgarsky [2019] |
| Mirror Descent (e.g. $\psi(\cdot) = \frac{1}{p} \left\| \cdot \right\|_p^p$) | $\mathrm{argmin}_w \left\| w - w_0 \right\|_p$ 
 s.t. $w$ fits all data 
 Gunasekar et al. [2018] 
 Azizan and Hassibi [2019a] | $\mathrm{argmin}_w \left\| w \right\|_p$ 
 s.t. $w$ classifies all data 

 **This work** |

used and it attains its minimum at a finite value. For example, the implicit bias of gradient descent (GD) for square loss goes back to Engl et al. [1996]. Beyond GD, analysis of other popular algorithms such as the family of mirror descent (MD), which is an important generalization of GD, is more involved and was established only recently by [Azizan and Hassibi, 2019a, Gunasekar et al., 2018]. Specifically, those works showed that mirror descent converges to the interpolating solution that is closest to the initialization in terms of a Bregman divergence. Thus, the implicit bias in linear regression is relatively well-understood by now.

On the other hand, **in the classification setting, the implicit bias analysis becomes significantly more challenging, and several questions remain open** despite significant progress in the past few years. A key differentiating factor in the classification setting is that the loss function does not attain its minimum at a finite value, and the weights have to grow to infinity. It has been shown that for the logistics loss, the gradient descent iterates converge to the $\ell_2$-maximum margin SVM solution in direction [Ji and Telgarsky, 2019, Soudry et al., 2018]. However, such characterizations for mirror descent are missing in the literature. Because it is possible for optimization algorithms to exhibit implicit bias in regression but not in classification (and vice versa) [Gunasekar et al., 2018], resolving this gap of knowledge warrants careful analysis. See Table 1 for a summary.

In this paper, we advance the understanding of the implicit regularization of mirror descent in the classification setting. In particular, inspired by their practicality, we focus on mirror descents with potential function $\psi(\cdot) = \frac{1}{p} \left\| \cdot \right\|_p^p$ for $p > 1$. More specifically, such choice of potential results in an update rule that *can be applied coordinate-wise*, in the sense that updating the value at one coordinate does not depend on the values at other coordinates. Thanks to this property, this subclass of mirror descent can be implemented with *no additional computational overhead*, making it much more practical than other algorithms in the literature; see Remark 10 for more details.

**Our contributions.** In this paper, we make the following contributions:

- We study mirror descent with potential $\frac{1}{p} \left\| \cdot \right\|_p^p$ for $p > 1$, which will call *p-norm GD*, and abbreviated as $p$-GD, as a practical and efficient generalization of the popular gradient descent.
- We show that for separable linear classification with logistics loss, $p$-GD exhibits implicit regularization by converging in direction to a "generalized" maximum-margin solution with respect to the $\ell_p$ norm. More generally, we show that, for monotonically decreasing loss functions, $p$-GD follows the so-called regularization path, which is defined in Section 2.
- We investigate the implications of our theoretical findings with two sets of experiments: Our experiments involving linear models corroborate our theoretical results, and real-world experiments with deep neural networks and popular datasets suggest that our findings carry over to such

nonlinear settings. Our deep learning experiments further show that $p$-GD with different $p$ lead to significantly different generalization performance.

**Additional related work.** We remark that recent works also attempt to accelerate the convergence of gradient descent to its implicit regularization, either by using an aggressive step size schedule [Ji and Telgarsky, 2021, Nacson et al., 2019] or with momentum [Ji et al., 2021]. Further, there have been several results for other optimization methods, including steepest descent, AdaBoost, and various adaptive methods such as RMSProp and Adam [Gunasekar et al., 2018, Min et al., 2022, Rosset et al., 2004, Telgarsky, 2013, Wang et al., 2021]. A mirror-descent-based algorithm for explicit regularization was recently proposed by Azizan et al. [2021a]. Comparatively, there has been very little progress on mirror descent in the classification setting. Li et al. [2021] consider a mirror descent, but their assumptions are not applicable beyond the $\ell_2$ geometry.[1] To the best of our knowledge, there is no result for more general mirror descent algorithms in the classification setting.

## 2    Background and Problem Setting

We are interested in the well-known classification setting. Consider a collection of input-label pairs $\{(x_i, y_i)\}_{i=1}^n \subset \mathbb{R}^d \times \{\pm 1\}$ and a classifier $f_w(x)$, where $w \in \mathcal{W}$. For some *loss function* $\ell : \mathbb{R} \times \{\pm 1\} \to \mathbb{R}$, our goal is to minimize the empirical loss:

$$L(w) = \frac{1}{n} \sum_{i=1}^n \ell(y_i \cdot f_w(x_i)).$$

Throughout the paper, we assume that the classification loss function $\ell$ is decreasing, convex and does not attain its minimum, as in most common loss functions in practice (e.g., logistics loss and exponential loss). Without loss of generality, we assume that $\inf \ell(\cdot) = 0$.

For our theoretical analysis, we consider a linear model, where the models can be expressed by $f_w(x) = w^\top x$ and $w \in \mathbb{R}^d$. We also make the following assumptions about the data. First, since we are mainly interested in the over-parameterized setting where $d > n$, we assume that the data is linearly separable, i.e., there exists $w^* \in \mathbb{R}^d$ s.t. $\text{sign}(\langle w^*, x_i \rangle) = y_i$ for all $i \in [n]$. We also assume that the inputs $x_i$'s are bounded. More specifically, for our later purpose, we assume that for $p > 0$, there exists some constant $C$ so that $\max_i \|x_i\|_q < C$, where $1/q + 1/p = 1$.

**Preliminaries on mirror descent.** The key component of mirror descent is a *potential function*. In this work, we will focus on differentiable and strictly convex potentials defined on the entire domain $\mathbb{R}^n$.[2] We call $\nabla \psi$ the corresponding *mirror map*. Given a potential, the natural notion of "distance" associated with the potential $\psi$ is given by the Bregman divergence.

**Definition 1** (Bregman divergence [Bregman, 1967])**.** *For a mirror map $\psi$, the Bregman divergence $D_\psi (\cdot, \cdot)$ associated to $\psi$ is defined as*

$$D_\psi (x, y) := \psi(x) - \psi(y) - \langle \nabla \psi(y), \ x - y \rangle, \qquad \forall x, y \in \mathbb{R}^n.$$

An important case is the potential $\psi = \frac{\|\cdot\|^2}{2}$, where $\|\cdot\|$ denotes the Euclidean norm. Then, the Bregman divergence becomes $D_\psi(x, y) = \frac{1}{2} \|x - y\|^2$. For more background on Bregman divergence and its properties, see, e.g., [Bauschke et al., 2017, Section 2.2] and [Azizan and Hassibi, 2019b].

Mirror descent (MD) with respect to the mirror map $\psi$ is a generalization of gradient descent where we use Bregman divergence as a measure of distance:

$$w_{t+1} = \operatorname*{argmin}_w \left\{ \frac{1}{\eta} D_\psi(w, w_t) + \langle \nabla L(w_t), \ w \rangle \right\} \tag{MD}$$

Equivalently, MD can be written as $\nabla \psi(w_{t+1}) = \nabla \psi(w_t) - \eta \nabla L(w_t)$. We refer readers to [Bubeck, 2015, Figure 4.1] for a nice illustration of mirror descent. Also, see [Juditsky et al., 2011, Section 5.7] for various examples of potentials depending on applications.

One property we will repeatedly use is the following [Azizan and Hassibi, 2019a]:

---

[1]To be precise, they assume that the Bregman divergence is lower and upper bounded by a constant factor of the squared Euclidean distance, e.g., as in the case of a squared Mahalanobis distance.

[2]In general, the mirror map is a convex function of Legendre type (see, e.g., [Rockafellar, 1970, Section 26]).

**Lemma 2** (MD identity). *For any $w \in \mathbb{R}^n$, the following identities hold for MD:*

$$D_\psi(w, w_t) = D_\psi(w, w_{t+1}) + D_{\psi-\eta L}(w_{t+1}, w_t) + \eta D_L(w, w_t) - \eta L(w) + \eta L(w_{t+1}), \quad (1a)$$

$$= D_\psi(w, w_{t+1}) + D_{\psi-\eta L}(w_{t+1}, w_t) - \eta \langle \nabla L(w_t), \ w - w_t \rangle - \eta L(w_t) + \eta L(w_{t+1}). \quad (1b)$$

Using Lemma 2, we make several new observations and prove the following useful statements.

**Lemma 3.** *For sufficiently small step size $\eta$ such that $\psi - \eta L$ is convex, the loss is monotonically decreasing after each iteration of MD, i.e., $L(w_{t+1}) \leq L(w_t)$.*

**Lemma 4.** *In a separable linear classification problem, if $\eta$ is chosen sufficiently small s.t. $\psi - \eta L$ is convex, then we have $L(w_t) \to 0$ as $t \to \infty$. Hence, $\lim_{t\to\infty} \|w_t\| = \infty$ for any norm $\|\cdot\|$.*

The formal proofs of these lemmas can be found in Appendix A.

**Remark 5.** We can relax the condition in Lemma 3 and 4 such that for a sufficiently small step size $\eta$, $\psi - \eta L$ is only locally convex at the iterates $w_t$. The relaxed condition allows us to analyze losses such as the exponential loss (see, e.g. Footnote 2 of Soudry et al. [2018]). This condition can be considered as the mirror descent counterpart to the standard smoothness assumption in the analysis of gradient descent (see Lu et al. [2018]).

**Preliminaries on the convergence of linear classifier.** As we discussed above, the weights vector $w_t$ diverges for mirror descent. Here the main theoretical question is:

What direction does MD diverge to? In other words, can we characterize $w_t / \|w_t\|$ as $t \to \infty$?

To answer this question, we define two special directions whose importance will be illustrated later.

**Definition 6.** *The **regularization path** with respect to the $\ell_p$-norm is defined as*

$$\bar{w}_p(B) = \underset{\|w\|_p \leq B}{\operatorname{argmin}} L(w) \quad (2)$$

*And if the limit $\lim_{B\to\infty} \bar{w}_p(B)/B$ exists, we call it the **regularized direction** and denote it by $u_p^r$.*

**Definition 7.** *The **margin** $\gamma$ of the a linear classifier $w$ is defined as $\gamma(w) = \min_{i=1,...,n} y_i \langle x_i, \ w \rangle$. The **max-margin direction** with respect to the $\ell_p$-norm is defined as:*

$$u_p^m := \underset{\|w\|_p \leq 1}{\operatorname{argmax}} \left\{ \min_{i=1,...,n} y_i \langle x_i, \ w \rangle \right\} \quad (3)$$

*And let $\hat{\gamma}_p$ be the optimal value to the equation above.*

Note that $u_p^m$ is parallel to the hard-margin SVM solution w.r.t. $\ell_p$-norm: $\operatorname{argmin}_w \{\|w\|_p : \gamma(w) \geq 1\}$. Also note that the superscripts in $u_p^r$ and $u_p^m$ are not variables and we only use this notation to differentiate the two definitions. Prior results had shown that, in linear classification, gradient descent converges in direction.

**Theorem 8** (Soudry et al. [2018]). *For separable linear classification with logistics loss, the gradient descent iterates with sufficiently small step size converge in direction to $u_2^m$, i.e., $\lim_{t\to\infty} \frac{w_t}{\|w_t\|_2} = u_2^m$.*

**Theorem 9** (Ji et al. [2020]). *If the regularized direction $u_p^r$ with respect to the $\ell_2$-norm exists, then the gradient descent iterates with sufficiently small step size converge to the regularized direction $u_2^r$, i.e., $\lim_{t\to\infty} \frac{w_t}{\|w_t\|_2} = u_2^r$.*

## 3 Mirror Descent with the $p$-th Power of $\ell_p$-norm

In this section, we investigate theoretical properties of the main algorithm of interest, namely mirror descent with $\psi(\cdot) = \frac{1}{p} \|\cdot\|_p^p$ and for $p > 1$.[3] We shall call this algorithm *p-norm GD* because it naturally generalizes gradient descent to $\ell_p$ geometry, and for conciseness, we will refer to this algorithm by the shorthand $p$-GD. As noticed by Azizan et al. [2021b], this choice of mirror potential

---

[3]Because the potential function must be *strictly* convex for Bregman divergence to be well-defined, the value of $p$ cannot be exactly 1.

is particularly of practical interest because the mirror map $\nabla \psi$ updates becomes *separable* in coordinates and thus can be implemented *coordinate-wise* independent of other coordinates:

$$\forall j \in [d], \quad \begin{cases} w_{t+1}[j] \leftarrow \left| w_t^+[j] \right|^{\frac{1}{p-1}} \cdot \text{sign}\left( w_t^+[j] \right) \\ w_t^+[j] := |w_t[j]|^{p-1} \text{sign}(w_t[j]) - \eta \nabla L(w_t)[j] \end{cases} \qquad (p\text{-}\mathsf{GD})$$

Furthermore, we can extend upon the observation of Azizan et al. [2021b] and derive these identities that allow us to better manipulate $p$-GD:

$$\langle \nabla \psi(w), \, w \rangle = \text{sign}(w_1)w_1 \cdot |w_1|^{p-1} + \cdots + \text{sign}(w_d)w_d \cdot |w_d|^{p-1} = \|w\|^p \qquad (4a)$$

$$D_\psi (cw, cw') = |c|^p D_\psi (w, w') \quad \forall c \in \mathbb{R}. \qquad (4b)$$

**Remark 10.** Note that the coordinate-wise separability property is not shared among other related algorithms in the literature. For instance, the choice $\psi = \frac{1}{2} \|\cdot\|_q^2$ for $1/p + 1/q = 1$, which is referred to as the $p$-norm algorithm [Gentile, 2003, Grove et al., 2001] is not fully coordinate-wise separable since it requires computing $\|w_t\|_p$ at each step (see, e.g., [Gentile, 2003, eq. (1)]). Another related algorithm is steepest descent, where the Bregman divergence in MD is replaced with $\|\cdot\|^2$ for general norm $\|\cdot\|$.[4] However, for similar reasons, the update rule is not fully separable.

## 3.1 Main theoretical results

We extend Theorems 8 and 9 to the setting of $p$-GD. We will resolve two major obstacles in the analysis of implicit regularization in linear classification:

- Our analysis approaches the classification setting as a limit of the regression implicit bias. This argument gives stronger theoretical justification for utilizing the regularized direction (as employed by Ji et al. [2020]) and partially addresses the concern from Gunasekar et al. [2018] that the implicit bias of regression and classification problems are "fundamentally different."
- On a more technical note, analyzing the implicit bias requires handling the cross terms of the form $\langle \nabla \psi(w), \, w' \rangle$, which lack direct geometric interpretations. We demonstrate that for our potential functions of interest, these terms can be nicely written and can be handled in the analysis.

We begin with the motivation behind the regularized direction, and consider the regression setting in which there exists some weight vector $w$ such that $L(w) = 0$. Then, we can apply Lemma 2 to get

$$D_\psi(w, w_t) = D_\psi(w, w_{t+1}) + D_{\psi - \eta L}(w_{t+1}, w_t) + \eta D_L(w, w_t) + \eta(L(w_{t+1}) - L(w))$$

Since we assumed $L(w) = 0$, the equation above implies that $D_\psi(w, w_t) \geq D_\psi(w, w_{t+1})$ for sufficiently small step-size $\eta$. This can be interpreted as MD having a decreasing "potential" of the from $D_\psi (w, \cdot)$ during each step. Using this property, Azizan and Hassibi [2019a] establishes the implicit bias results of mirror descent in the regression setting.

However, such weight vector $w$ does not exist in the classification setting. One natural workaround would then be to choose a vector $w$ so that $L(w) \leq L(w_t)$ for all $t \leq T$. The following result, which is a generalization of [Ji et al., 2020, Lemma 9], shows that one can in fact choose the reference vector $w$ as a scalar multiple of the regularized direction.

**Lemma 11.** *If the regularized direction $u_p^r$ exists, then $\forall \alpha > 0$, there exists $r_\alpha$ such that for any $w$ with $\|w\|_p > r_\alpha$, we have $L((1 + \alpha) \|w\|_p u_p^r) \leq L(w)$.*

However, this does not resolve the issue altogether. Recall from Lemma 4 that the loss approaches 0, and therefore one cannot choose a fixed reference vector $w$ in the limit as $T \to \infty$. But due to the homogeneity of Bregman divergence (4b), we can scale $u_p^r$ by a constant factor during each iteration, and, by doing so, we choose the reference vector $w$ to be a "moving target." In other words, the idea behind our analysis is that the classification problem is chasing after a regression one and would behave similar to it in the limit. Let us formalize this idea. We begin with the following inequality:

$$D_\psi \left( c_t u_p^r, w_{t+1} \right) \leq D_\psi \left( c_t u_p^r, w_t \right) - \eta L(w_{t+1}) + \eta L(w_t), \qquad (5)$$

---

[4]It is also worth noting that steepest descent is not an instance of mirror descent since $\|\cdot\|^2$ is not a Bregman divergence for a general norm $\|\cdot\|$.

where $c_t$ is taken to be $\approx \|w_t\|_p$.[5]

Now we modify (5) so that it can telescope over different iterations. One way is to add $D_\psi\left(c_{t+1}u_p^r, w_{t+1}\right)$ on both sides of (5) and move $D_\psi\left(c_t u_p^r, w_{t+1}\right)$ to the right-hand side as follows:

$$
\begin{aligned}
&D_\psi\left(c_{t+1}u_p^r, w_{t+1}\right) \\
&\leq D_\psi\left(c_t u_p^r, w_t\right) - \eta L(w_{t+1}) + \eta L(w_t) + D_\psi\left(c_{t+1}u_p^r, w_{t+1}\right) - D_\psi\left(c_t u_p^r, w_{t+1}\right) \\
&= D_\psi\left(c_t u_p^r, w_t\right) - \eta L(w_{t+1}) + \eta L(w_t) + \psi(c_{t+1}u_p^r) - \psi(c_t u_p^r) - \left\langle \nabla\psi(w_{t+1}),\ (c_{t+1}-c_t)u_p^r \right\rangle
\end{aligned}
$$

Summing over $t = 0, \ldots, T-1$ gives us

$$
\begin{aligned}
D_\psi\left(c_T u_p^r, w_T\right) \leq\ &D_\psi\left(c_0 u_p^r, w_0\right) - \eta L(w_1) + \eta L(w_T) + \psi(c_T u_p^r) - \psi(c_1 u_p^r) \\
&- \sum_{t=1}^{T-1} \left\langle \nabla\psi(w_{t+1}),\ (c_{t+1}-c_t)u_p^r \right\rangle
\end{aligned} \tag{6}
$$

The rest of the argument deals with simplifying quantities that do not cancel under telescoping sum. For instance, in order to deal with $\left\langle \nabla\psi(w_{t+1}),\ u_p^r \right\rangle$, we invoke the MD update rule as follows

$$
\left\langle \nabla\psi(w_{t+1}) - \nabla\psi(w_t),\ u_p^r \right\rangle = \left\langle -\eta\nabla L(w_t),\ u_p^r \right\rangle \gtrsim \left\langle -\eta\nabla L(w_t),\ w_t \right\rangle,
$$

where the last inequality follows from the intuition that $u_p^r$ is the direction along which the loss achieves the smallest value and hence $\nabla L(w_t)$ must point away from $u_p^r$, i.e., it must be that $\left\langle \nabla L(w_t),\ u_p^r \right\rangle \lesssim \left\langle \nabla L(w_t),\ u \right\rangle$ for any direction $u$. The following result formalizes this intuition.

**Corollary 12.** *For $w$ so that $\|w\|_p > r_\alpha$, we have $\left\langle \nabla L(w),\ w \right\rangle \geq (1+\alpha)\|w\|_p \left\langle \nabla L(w),\ u_p^r \right\rangle$.*

*Proof.* This follows from the convexity of $L$ and Lemma 11: $\left\langle \nabla L(w),\ w - (1+\alpha)\|w\|u_p^r) \right\rangle \geq L(w) - L((1+\alpha)\|w\|u_p^r) \geq 0$. $\square$

Now we are left with the terms $\left\langle -\eta\nabla L(w_t),\ w_t \right\rangle$. For general potential $\psi$, the quantity $\left\langle -\eta\nabla L(w_t),\ w_t \right\rangle = \left\langle \nabla\psi(w_{t+1}) - \nabla\psi(w_t),\ w_t \right\rangle$ cannot be simplified. On the other hand, due to our choice of potential, one can invoke Lemma 2 to lower bound these quantities in terms of $\|w_{t+1}\|_p$ and $\|w_t\|_p$, and this step is detailed in Lemma 18 in Appendix B.2. Once we have established a lower bound on $\left\langle \nabla\psi(w_{t+1}),\ u_p^r \right\rangle$, we can turn (6) entirely into a telescoping sum and unwind the above process to show that $D_\psi\left(u_p^r, w_t/\|w_t\|_p\right)$ must converge to zero in the limit as $t \to \infty$. Putting this all together, we obtain the following result.

**Theorem 13.** *For a separable linear classification problem, if the regularized direction $u_p^r$ exists, then with sufficiently small step size, the iterates of $p$-GD converge to $u_p^r$ in direction:*

$$
\lim_{t\to\infty} \frac{w_t}{\|w_t\|_p} = u_p^r. \tag{7}
$$

A formal proof of this theorem can be found in Appendix B.3. We note that our proof further simplifies derivations using the separability of the mirror map. The final missing piece would be the existence of the regularized direction. In general, finding the limit direction $u_p^r$ would be difficult. Fortunately, we can sometimes appeal to the max-margin direction that is much easier to compute. The following result is a generalization of [Ji et al., 2020, Proposition 10] and shows that for common losses in classification, the regularized direction and the max-margin direction are the same, hence proving the existence of the former.

**Proposition 14.** *If we have a loss with exponential tail, e.g. $\lim_{z\to\infty} \ell(z)e^{az} = b$, then the regularized direction exists and it is equal to the max-margin direction $u_p^m$.*

The proof of this result can be found in Appendix B.5. Note that many commonly used losses in classification, e.g., logistic loss, have exponential tail.

---

[5]To be more precise, we want $c_t = (1+\alpha)\|w_t\|_p$; and reason behind this choice is self-evident after we present Corollary 12.

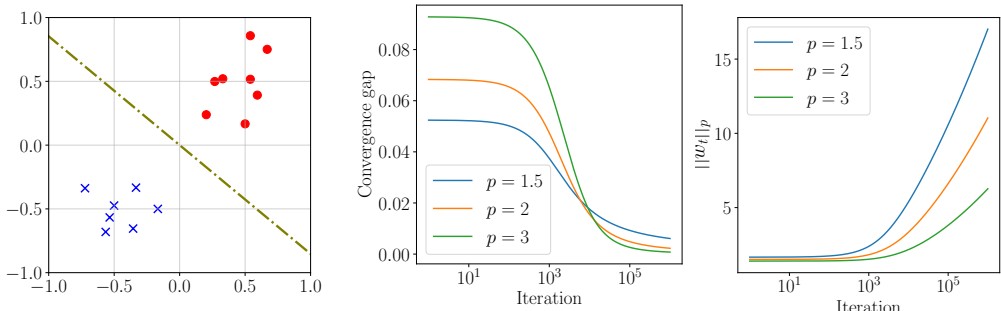

Figure 1: An example of $p$-GD on randomly generated data with exponential loss and $p = 1.5, 2, 3$.
**(1)** The left plot is a scatter plot of the data: $\times$'s and $\bullet$'s denote the two different labels ($y_i = \pm 1$). The dotted line is the $\ell_2$ max-margin classifier. For clarity, other $\ell_p$ max-margin classifiers are omitted from the plot. **(2)** The middle plot shows the rate which the quantity $D_\psi \left( u_p^r, w_t / \|w_t\|_t \right)$ converges to 0. **(3)** The right plot shows how fast the $p$-norm of $w_t$ growths. We can observe that the asymptotic behaviors of these plots are consistent with Corollary 17.

## 3.2 Asymptotic convergence rate

In this section, we characterize the rate of convergence in Theorem 13. Following the proof of Theorem 13, one can show the following result in the case of linearly separable data.

**Corollary 15.** *The following rate of convergence holds:*

$$D_\psi \left( u_p^r, \frac{w_t}{\|w_t\|_p} \right) \in O \left( \|w_t\|_p^{-(p-1)} \right).$$

In order to fully understand the convergence rate, we need to characterize the asymptotic behavior of $\|w_t\|_p$. The next result precisely does that. Recall that we assumed the dataset is bounded so that $\max_i \|x_i\|_q \leq C$ for $1/p + 1/q = 1$, and the max-margin direction $u_p^m$ satisfies $\langle x_i, u_p^m \rangle \geq \hat{\gamma}_p \ \forall i \in [n]$. Then, we have the following bound on $\|w_t\|_p$.

**Lemma 16.** *For exponential loss $\ell(z) = \exp(-z)$, the asymptotic growth of $\|w_t\|_p$ is contained in $\Theta(\log t)$. In particular, we have*

$$\liminf_{t \to \infty} \|w_t\|_p \geq \frac{1}{C} (\log t - p \log \log t) + O(1) \ and \ \limsup_{t \to \infty} \|w_t\|_p \leq \hat{\gamma}_p^{-1} \frac{p}{p-1} \log t.$$

The proof of this lemma can be found in Appendix C. Similar conclusions can be reached for other losses with exponential tail. Therefore, in such cases, $p$-GD has poly-logarithmic rate of convergence.

**Corollary 17.** *For exponential loss, we have convergence rate*

$$D_\psi \left( u_p^r, \frac{w_t}{\|w_t\|_p} \right) \in O \left( \frac{1}{\log^{p-1}(t)} \right).$$

# 4 Experiments

In this section, we investigate the behavior and performance of $p$-GD for various values of $p$. We naturally pick $p = 2$ that corresponds to gradient descent, Because $p$-GD does not directly support $p = 1$ and $\infty$, we choose $p = 1.1$ as a surrogate for $\ell_1$, and $p = 10$ as a surrogate for $\ell_\infty$. We also consider $p = 1.5, 3, 6$ to interpolate these points. This section will present a summary of our results; the complete experimental setup and full results can be found in Appendices E and F.

## 4.1 Linear classification

**Visualization of the convergence of $p$-GD.** To visualize the results of Theorem 13 and Corollary 17, we randomly generated a linearly separable set of 15 points in $\mathbb{R}^2$. We then employed $p$-GD on this

dataset with exponential loss $\ell(z) = \exp(-z)$ and fixed step size $\eta = 10^{-4}$. We ran this experiment for $p = 1.5, 2, 3$ and for $10^6$ iterations.

In the illustrations of Figure 1, the mirror descent iterates $w_t$ have unbounded norm and converge in direction to $u_p^{\mathsf{m}}$. These results are consistent with Lemma 4 and with Theorem 13. Moreover, as predicted by Corollary 17, the exact rate of convergence for $D_\psi \left( u_p^{\mathsf{m}}, w_t / \|w_t\|_t \right)$ is poly-logarithmic with respect to the number of iterations. Corollary 17 also indicates that the convergence rate would be faster for larger $p$ due to the larger exponent, and this is consistent with our observation in the second plot of Figure 1. Finally, in the third plot of Figure 1, the norm of the iterates $w_t$ grows at a logarithmic rate, which is the same as the prediction by Lemma 16.

**Implicit bias of $p$-GD in linear classification.** We now verify the conclusions of Theorem 13. To this end, we recall that $u_p^{\mathsf{m}}$ is parallel to the SVM solution $\operatorname{argmin}_w \{\|w\|_p : \gamma(w) \geq 1\}$. Hence, we can exploit the linearity and rescale any classifier so that its margin is equal to 1. If the prediction of Theorem 13 holds, then for each fixed value of $p$, the classifier generated by $p$-GD should have the smallest $\ell_p$-norm after rescaling.

To ensure that $u_p^{\mathsf{m}}$ are sufficiently different for different values of $p$, we simulate an over-parameterized setting by randomly select 15 points in $\mathbb{R}^{100}$. We used fixed step size of $10^{-4}$ and ran 250 thousand iterations for different $p$'s.

Table 2 shows the results for $p = 1.1, 2, 3$ and 10; under each norm, we highlight the smallest classifier in **boldface**. Among the four classifiers we presented, $p$-GD with $p = 1.1$ has the smallest $\ell_{1.1}$-norm. And similar conclusions hold for $p = 2, 3, 10$. Although $p$-GD converges to $u_p^{\mathsf{m}}$ at a very slow rate, we are able to observe a very strong implicit bias of $p$-GD classifiers toward their respective $\ell_p$ geometry in a highly over-parameterized setting. This suggests we should be able to take advantage of the implicit regularization in practice and at a moderate computational cost. Due to space constraints, we defer a more complete result with additional values of $p$ to Appendix F.1.

## 4.2 Deep neural networks

Going beyond linear models, we now investigate $p$-GD in deep-learning settings in its impact on the structure of the learned model and potential implications on the generalization performance. As we had discussed in Section 3, **the implementation of $p$-GD is straightforward**; to il-

Table 2: Size of the linear classifiers generated by $p$-GD (after rescaling) in $\ell_{1.1}, \ell_2, \ell_3$ and $\ell_{10}$ norms.

|  | $\ell_{1.1}$ | $\ell_2$ | $\ell_3$ | $\ell_{10}$ |
|---|---|---|---|---|
| $p = 1.1$ | **5.670** | 1.659 | 1.100 | 0.698 |
| $p = 2$ | 6.447 | **1.273** | 0.710 | 0.393 |
| $p = 3$ | 7.618 | 1.345 | **0.691** | 0.318 |
| $p = 10$ | 9.086 | 1.520 | 0.742 | **0.281** |

lustrate simplicity of implementation, we provide code snippets in Appendix D. Thus, we are able to effectively experiment with the behaviors $p$-GD in neural network training. Specifically, we perform a set of experiments on the CIFAR-10 dataset [Krizhevsky et al., 2009]. We use the *stochastic* version of $p$-GD with different values of $p$. We choose a variety of networks: VGG [Simonyan and Zisserman, 2014], RESNET [He et al., 2016], MOBILENET [Sandler et al., 2018] and REGNET [Radosavovic et al., 2020].

**Implicit bias of $p$-GD in deep neural networks.** Since the notion of margin is not well-defined in this highly nonlinear setting, we instead visualize the impacts of $p$-GD's implicit regularization on the histogram of weights (in absolute value) in the trained model.

In Figure 2, we report the weight histograms of RESNET-18 models trained under $p$-GD with $p = 1.1, 2, 3$ and 10. Depending on $p$, we observe interesting differences between the histograms. Note that the deep network is most sparse when $p = 1.1$ as most weights clustered around 0. Moreover, comparing the maximum weights, one can see that the case of $p = 10$ achieves the smallest value. Another observation is that the network becomes denser as $p$ increases; for instance, there are more weights away from zero for the cases $p = 3, 10$. These overall tendencies are also observed for other deep neural networks; see Appendix F.2.

**Generalization performance.** We next investigate the generalization performance of networks trained with different $p$'s. To this end, we adopt a fixed selection of hyper-parameters and then train four deep neural network models to 100% training accuracy with $p$-GD with different $p$'s. As

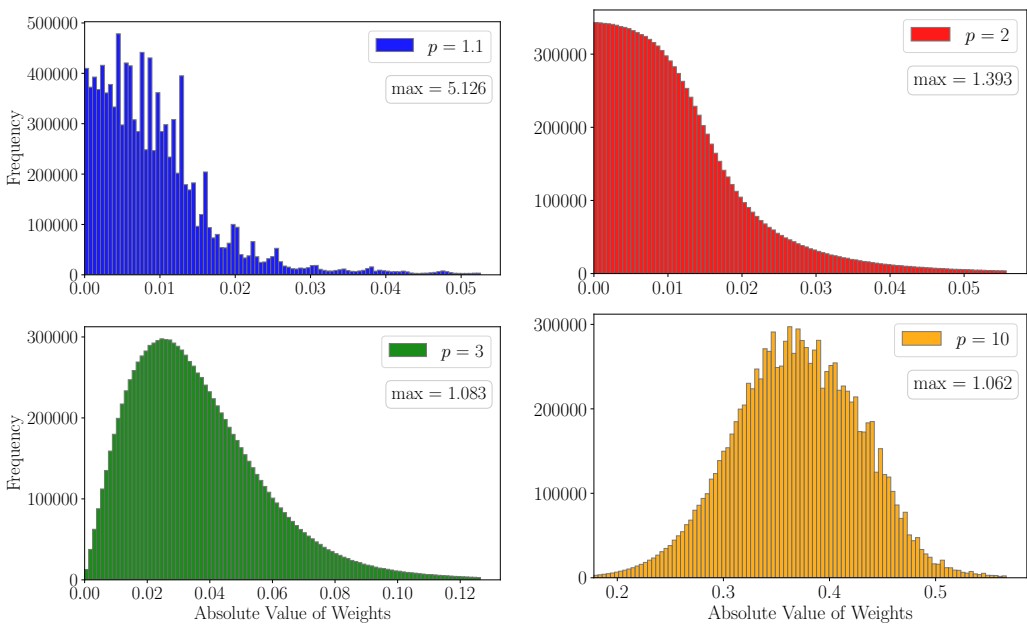

Figure 2: The histogram of weights in RESNET-18 models trained with $p$-GD for the CIFAR-10 dataset. For clarity, we cropped out the tails and each plot has 100 bins after cropping. The trends of these histograms reflect the implicit biases of $p$-GD: the distribution of $p = 1.1$ has the most number of weights around zero; and the maximum weight is smallest when $p = 10$.

Table 3: CIFAR-10 test accuracy (%) of $p$-GD on various deep neural networks. For each deep network and value of $p$, the average $\pm$ std. dev. over 5 trials are reported. And the best performing value(s) of $p$ for each individual deep network is highlighted in **boldface**.

|  | VGG-11 | RESNET-18 | MOBILENET-V2 | REGNETX-200MF |
|---|---|---|---|---|
| $p = 1.1$ | $88.19 \pm .17$ | $92.63 \pm .12$ | $91.16 \pm .09$ | $91.21 \pm .18$ |
| $p = 2$ (SGD) | $90.15 \pm .16$ | $\mathbf{93.90} \pm .14$ | $91.97 \pm .10$ | $92.75 \pm .13$ |
| $p = 3$ | $\mathbf{90.85} \pm .15$ | $\mathbf{94.01} \pm .13$ | $\mathbf{93.23} \pm .26$ | $\mathbf{94.07} \pm .12$ |
| $p = 10$ | $88.78 \pm .37$ | $93.55 \pm .21$ | $92.60 \pm .22$ | $92.97 \pm .16$ |

Table 3 shows, interestingly the networks trained by $p$-GD with $p = 3$ consistently outperform other choices of $p$'s; notably, for MOBILENET and REGNET, the case of $p = 3$ outperforms the others by more than 1%. Somewhat counter-intuitively, the sparser network trained by $p$-GD with $p = 1.1$ does not exhibit better generalization performance, but rather shows worse generalization than other values of $p$. Although these observations are not directly predicted by our theoretical results, we believe that they nevertheless establish an important step toward understanding generalization of overparameterized models. Due to space limit, we defer other experimental results to Appendix F.3.

**IMAGENET experiments.** We also perform a similar set of experiments on the IMAGENET dataset [Russakovsky et al., 2015], and these results can be found in Appendix F.4.

## 5  Conclusion and Future Work

In this paper, we establish an important step towards better understanding implicit bias in the classification setting, by showing that $p$-GD converges in direction to the generalized regularized/max-margin directions. We also run several experiments to corroborate our main findings along with the practicality of $p$-GD. The experiments are conducted in various settings: (i) linear models in both low and high dimensions, (ii) real-world data with highly over-parameterized nonlinear models.

We conclude this paper with several important future directions:

- Our analysis holds for $\psi(\cdot) = \|\cdot\|_p^p$, where we argued that this choice is key practical interest due to its efficient algorithmic implementations. It is mathematically interesting to generalize our analysis to other potential functions regardless of practical interest.
- As we discussed in Section 4.2, different choices of $p$'s for our $p$-GD algorithm result in different generalization performance. It would be interesting to investigate this phenomenon and to develop theory that explains why certain values of $p$ lead to better generalization performance.
- Another interesting question is to further investigate how practical techniques used in training neural networks (such as weight decay and adaptive learning rate) can affect the implicit bias and generalization properties of $p$-GD.

## Acknowledgement

The authors thank MIT UROP students Tiffany Huang and Haimoshri Das for contributing to the experiments in Section 4.2. The authors acknowledge the MIT SuperCloud and Lincoln Laboratory Supercomputing Center for providing computing resources that have contributed to the results reported within this paper. This work was supported in part by MathWorks and the MIT-IBM Watson AI Lab. K.A. acknowledges support through graduate assistantship in part from the NSF grant 1846088, the ONR grant N00014-20-1-2394, and the VBFF. N.A. also acknowledges support from the Edgerton Career Development Professorship.

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
