# OpenReview forum: "Mirror Descent Maximizes Generalized Margin and Can Be Implemented Efficiently"
_NeurIPS.cc/2022/Conference — NeurIPS 2022 Accept_

### Official Review · Reviewer_UoZK · 2022-06-20

**Rating:** 6
**Confidence:** 3
**Soundness:** 3 good
**Presentation:** 2 fair
**Contribution:** 2 fair

**Summary:**

This paper studies theoretical details of convergence point of Mirror Descent algorithm those are having potential function as p-th power of l_p norm. Moreover, in the case of linearly separable classification the convergence has been shown to be in the direction which is called generalized maximum margin.

**Questions:**

- To me it seems that if we check multiple values of p, one of them likely to overperform p=2 case (SGD), so i was curious if there is way to do this process better. Also, it would be nice to have accuracy studies in experimentation with other Optimization Algorithms.

- Is it possible to understand characteristics of p-GD solutions with more complicated decision boundaries ?



**Ethics Review Area:**

["I don’t know"]

**Limitations:**

-

**Strengths And Weaknesses:**

Strengths:

- Paper is theoretically sound, and investigates theoretical characteristics of MD with more suitable potential function 1/p || . ||_p^{p} which could be applicable to train Deep Learning trainings.
- The paper studies further characteristics of linearly separable classes which is an important first step question to answer.
- There has been experiments performed to verify results.


Weaknesses:

- This would be interpreted as question as well instead of weakness, how do you choose the best p depending on the problem. Is this a new hyper-parameter to tune or there is a structured method to find the necessary p. I am more pointing this issue in terms of experimentation ?
-  Studying characteristics of linearly separable problems is interesting, but is there any way that p-GD could be studied more complicated decision boundaries ?

---

> ### Author Response · Authors · 2022-08-02
> **Author reponse to Reviewer UoZK**
>
> Thank you for your valuable comments and feedback.
>
> At a high level, you are correct that $p$ can be treated as another hyperparameter. However, we note that one may be able to choose it in a more clever way than through a grid search. In particular, if we have some knowledge about the underlying geometry of the problem, we can pick appropriate values of $p$ as prescribed by our theory. Alternatively, if we desire a certain quality from the learned classifier, e.g., a sparse network or weights with a small dynamic range, we can pick $p$ to induce such properties in the classifier (i.e., $p$ close to 1 or large, respectively).
>
> We understand that the setting of linear classification may seem restrictive. However, this is already challenging and is a standard setting in the study of implicit regularization, see [1-5]. Additionally, to the best of our knowledge, even in the well-studied case of gradient descent, the current state-of-art analysis is concerned with relatively simple models such as two-layer neural networks or linear networks without activation functions, see [6-8], which still require significant effort.
>
> [1] Daniel Soudry, Elad Hoffer, Mor Shpigel Nacson, Suriya Gunasekar, and Nathan Srebro. The implicit bias of gradient descent on separable data. The Journal of Machine Learning Research, 2018.
>
> [2] Ziwei Ji, and Matus Telgarsky. "The implicit bias of gradient descent on nonseparable data." In Conference on Learning Theory, pp. 1772-1798. PMLR, 2019.
>
> [3] Suriya Gunasekar, Jason Lee, Daniel Soudry, and Nathan Srebro. "Characterizing implicit bias in terms of optimization geometry." In International Conference on Machine Learning. PMLR, 2018.
>
> [4] Muthukumar, Vidya, Adhyyan Narang, Vignesh Subramanian, Mikhail Belkin, Daniel Hsu, and Anant Sahai. "Classification vs regression in overparameterized regimes: Does the loss function matter?." The Journal of Machine Learning Research, 2021.
>
> [5] Ziwei Ji, Nathan Srebro, and Matus Telgarsky. "Fast margin maximization via dual acceleration." In International Conference on Machine Learning. PMLR, 2021.
>
> [6] Suriya Gunasekar, Jason D. Lee, Daniel Soudry, and Nati Srebro. "Implicit bias of gradient descent on linear convolutional networks." Advances in Neural Information Processing Systems, 2018.
>
> [7] Lenaic Chizat, and Francis Bach. "Implicit bias of gradient descent for wide two-layer neural networks trained with the logistic loss." In Conference on Learning Theory. PMLR, 2020.
>
> [8] Gal Vardi, and Ohad Shamir. "Implicit regularization in ReLU networks with the square loss." In Conference on Learning Theory. PMLR, 2021.

---

> > ### Comment · Reviewer_UoZK · 2022-08-08
> > **Thanks**
> >
> > Thanks for the answers and clarifications. As a personal preference i would still like to see a structured study regarding choosing algorithm of p, or at least some case study, but this wouldn't stop me from increasing my score to weak accept.
> >
> > Thanks for the detailed response !

---

> > > ### Author Response · Authors · 2022-08-09
> > > **Thank you**
> > >
> > > Thank you again for your comments and for increasing your score. We agree that a deeper understanding of the effect of $p$ on generalization performance would be an interesting future work. We have incorporated this into Section 5 in the new version.

---

### Official Review · Reviewer_ZhPm · 2022-06-23

**Rating:** 7
**Confidence:** 4
**Soundness:** 3 good
**Presentation:** 3 good
**Contribution:** 3 good

**Summary:**

This work studies mirror descent in the classification setting with exponential and logistic losses and $p$-norm (to power $p$) potential functions, proving that when data is linearly separable and the step size is sufficiently small, this method converges to the maximum margin direction under the $p$-norm. The authors also describe rates of convergence as a function of iteration. Lastly, the authors perform numerical experiments demonstrating the claims of the theory for toy linear classification problems, as well as application to common deep network models on CIFAR-10, demonstrating the resulting weights have a distribution that reflect the structure imposed by the implicit norm regularization.

**Questions:**

I have one major question for the authors below. I assume that they will answer it satisfactorily, so I have decided to tentatively give this paper a **weak accept**. Depending on the response, I will increase my score to **accept** or decrease to **reject**.

**Questions:**
- The authors appear to be somewhat inconsistent regarding which loss function each result applies for. At the start of the paper (line 84), they suggest that most results will apply for exponential loss and logistic loss. However, it appears that the exponential loss cannot satisfy Lemma 2, because $\psi - \eta \exp(\cdot)$ can never be convex. On the other hand, the results on convergence rate are given only for the exponential loss (with proof for the logistic loss left as an exercise for the reader). Can the authors either 1) fix Lemma 2 so that the exponential loss works, or 2) specialize all results to the logistic loss, and omit the exponential loss in result statements?

**Suggestions:**
- Point out that when $d > n$, linear separability of the data is trivially satisfied. Additionally, comment more on the importance of the separability assumption (allowing interpolation / the loss to going to zero).
- Make a note that while the hinge loss does not satisfy the assumptions (it attains its minimum), it gives the same maximum-margin solution.
- More clearly define $u_p^\mathsf{m}$ and $u_p^\mathsf{r}$, and make it clear that the superscripts $\mathsf{m}$ and $\mathsf{r}$ are not parameters of the problem such as sample sizes.
- Explain why the choice of $p = 1$ doesn't work.
- Add the citation [1] discussed under *Significance* above.

**Formatting/typographical:**
- Use a unified numbering scheme for theorems/remarks/definitions/etc. With so many, it is difficult to find them.
- (minor) Tables: use `booktabs` package
- line 206: "has exponential tail" -> "have exponential tail"
- line 251: "space constraint" -> "space constraints"

***

**Edit after review responses**

The authors have satisfactorily responded to my main question, so I have increased my score from **weak accept** to **accept**.

**Limitations:**

As mentioned above, the analysis as-is appears to only apply to the logistic loss, even though it is presented in a generic manner that would appear to be able to accommodate other common losses like exponential and hinge. I found no ethical limitations with this work.

**Strengths And Weaknesses:**

**Originality:**
This work builds on a rich literature of previous work in theoretical analysis of implicit regularization and mirror descent. The main theoretical contribution appears to be that the authors have extended the analysis of Azizan et al. on mirror descent to the classification case where the loss is minimized only for an infinite-norm solution. While Gunasekar et al. have studied classification previously under exponential loss with steepest descent, this work analyzes mirror descent with separable update rules for the logistic loss, which is a more practically relevant setting. I think this work could do a better job of contrasting itself with previous analysis

**Quality & Clarity:**
I believe the theoretical results presented to be technically sound. The biggest issue is that there appears to be some inconsistency with respect to the loss function from result to result, which is a bit confusing to draw conclusions from (I elaborate in the *Questions* section).

**Significance:**
It is of critical importance that we understand the implicit biases of our optimization algorithms in this era of overparameterized machine learning, and this work makes an important contribution by demonstrating how $p$-norm regularization can be achieved solely via efficient optimization methods in classification problems. In particular, it has been recently shown [1] that fast rates for interpolation in noisy settings require $p < 2$, so it is crucial that we go beyond standard gradient descent, both in theory and in practice. This work is an important step in that direction.

[1] Donhauser et al., "Fast rates for noisy interpolation require rethinking the effects of inductive bias." https://arxiv.org/abs/2203.03597

---

> ### Author Response · Authors · 2022-08-02
> **Author response to Reviewer ZhPm**
>
>
> We thank the reviewer for their valuable comments and suggestions.
>
> Let us first address the reviewer’s main concern regarding Lemma 2 (Lemma 3 in the revision). It is true that the exponential loss is not globally smooth whereas the logistics loss has the global smoothness property. However, we’d like to highlight that Lemma 2 only relies on local smoothness at the iterates, and this observation has been made in the literature for the GD case ($p=2$) as well [1, 2]. Below, we briefly explain how this is addressed and argue that the same argument can be applied to our setting as well.
>
> The key observation is that the exponential loss function is not smooth only when the loss is arbitrarily large. Therefore, based on the value of the loss at the initial iterate, we can upper bound the smoothness of the exponential loss function for all points that have a lower loss value than the initial point. Then, we can apply Lemma 2 to all the iterates of mirror descent. A similar argument had been raised in footnote 2 of [1]. Therefore, the exponential loss function fits under our framework. *And therefore our analysis is general in the sense that it applies to a variety of common loss functions.* To address the reviewer’s concern, we have further clarified this point in our first Remark.
>
>
> Regarding the reviewer’s suggestions:
> - We made additional remarks about the separability assumption where we introduced our problem setting at line 88.
> - Regarding the comment about the hinge loss, as noted by the reviewer, it attains its minimum at a finite point and does not satisfy the assumption of monotonically decreasing loss. Therefore, our analysis does not extend to this case, similar as in prior work in the literature [1, 2, 3]. However, the reviewer does raise an interesting point and we would like to investigate this question if it has not already been answered in literature.
> - We used the notation $u^\textsf{r}_p$ and $u^\textsf{m}_p$ because we feel that the alternative of using hats and overlines is less aesthetically pleasing. We chose to use a different font for the superscripts to reduce the chance of any confusion. If the reviewer deems a different notation more appropriate, we would be happy to change that.
> - 1-norm is not strictly convex and therefore cannot be used to define a Bregman divergence. This is why we restrict our analysis to the case where $p > 1$.
> - We appreciate the relevant reference mentioned and have added that in the revised version.
>
> Regarding formatting:
> - We have updated the numbering of theorems/definitions/etc as the reviewer recommended.
> - We thank the reviewer for pointing out the typos, and we have corrected them in our revision.
>
> Re limitations: We want to reiterate that, as discussed above, our analysis is quite general and applies to both exponential and logistics losses.
>
> [1] Daniel Soudry, Elad Hoffer, Mor Shpigel Nacson, Suriya Gunasekar, and Nathan Srebro. The implicit bias of gradient descent on separable data. The Journal of Machine Learning Research, 2018.
>
> [2] Ziwei Ji, Miroslav Dudík, Robert E. Schapire, and Matus Telgarsky. "Gradient descent follows the regularization path for general losses." In Conference on Learning Theory. PMLR, 2020.
>
> [3] Suriya Gunasekar, Jason Lee, Daniel Soudry, and Nathan Srebro. "Characterizing implicit bias in terms of optimization geometry." In International Conference on Machine Learning. PMLR, 2018.

---

> > ### Comment · Reviewer_ZhPm · 2022-08-03
> > **Satisfactory response**
> >
> > I find the authors' response and revision regarding the smoothness of the exponential loss to be satisfactory, so as promised, I will increase my score to **accept**.
> >
> > I am glad to see that the authors have already incorporated some of my suggestions, and I hope that they will incorporate the remainder of them in the camera-ready revision, as I believe that they will improve the clarity of this paper.

---

> > > ### Author Response · Authors · 2022-08-09
> > > **Thank you**
> > >
> > > Thank you again for your valuable feedback and for increasing your score. We have integrated the remaining suggestions into the paper.

---

### Official Review · Reviewer_FypC · 2022-07-02

**Rating:** 6
**Confidence:** 4
**Soundness:** 3 good
**Presentation:** 3 good
**Contribution:** 3 good

**Summary:**

This paper derives the implicit regularization effect for the mirror descent algorithm for linear classification tasks with the exponential-tailed loss and separable data. Specifically, with the potential function being $\Vert * \Vert^p_p$, it is shown that mirror descent converges to the $\ell^p$ max-margin solution. Experiments over linear models and deep neural networks are conducted to support the theoretical findings.

**Questions:**

1. More discussion on the importance of mirror descent in model deep learning is appreciated.
2. In [Soudry et al., 2017], a synthetic dataset is constructed with its max-margin solution known in prior, and the convergence to max-margin solution can thus be verified directly. It is suggested that the authors apply this methodology to verify the correctness of the theoretical results.
3. In this paper, special potential functions are considered. What is the difficulty to extend the results to general potential functions?

**Limitations:**

Please refer to the "weakness".

**Strengths And Weaknesses:**

Strengths:
1. This paper is well-written and I enjoy reading the paper.
2. The theoretical analysis is solid and explicitly explained (the connection with [Ji et al., 2020] is clearly pointed out). The experiment is detailed.

Weakness:
The major weakness lies in the motivation of the paper. The implicit bias of gradient descent (with momentum) and adaptive first-order optimizers over classification problems [Soudry et al., 2017; Qian et al. 2019; Wang et al., 2021] are studied due to their popularity in deep learning's application over classification tasks. However, mirror descent is less employed. Therefore, while it is good to know such a result, this problem has limited importance.

Reference:
1. Ji et al. Gradient descent follows the regularization path for general losses, 2020
2. Soudry et al. The Implicit Bias of Gradient Descent on Separable Data, 2017
3. Qian et al. The Implicit Bias of AdaGrad on Separable Data, 2019
4. Wang et al. Does Momentum Change the Implicit Regularization on Separable Data, 2021

---

> ### Author Response · Authors · 2022-08-02
> **Author response to Reviewer FypC**
>
> We thank the reviewer for the valuable comments and feedback.
>
> We would like to first clarify that one of the goals of our paper is to demonstrate the practicality of $p$-GD for deep learning. While mirror descent is a general family of optimization algorithms, it has not been employed in deep learning (other than GD, of course). The primary reason for that is the MD update rule requires applying the inverse of the mirror map, which is computationally expensive, especially for deep neural networks with many parameters. However, the significance of the $p$-GD class of mirror descent for deep learning is that (1) the update rule for $p$-GD becomes fully parallelizable and thus efficiently implementable in the same manner as GD; (2) different $p$ lead to weight vectors with significantly different geometries, and our experiments in Section 4.2 demonstrate that this leads to different generalization performance.
>
> Regarding the synthetic experiment from Soudry et al.: This is exactly what we have done. More specifically, inspired by the experiments in Soudry et al. [1] and Nacson et al. [2], we conducted a set of synthetic experiments in Section 4.1 to verify the convergence to the (generalized) max-margin solution, and those results corroborate with our main theoretical results.
>
> The biggest challenge in extending the result to a more general potential function is the homogeneity property. With our choice of potential function, if we scale two vectors by the same constants, then their Bregman divergence is also scaled by a constant. However, this property no longer holds for general potential function and therefore it is unclear how to deal with vector normalizations in such cases.
>
>
> [1] Daniel Soudry, Elad Hoffer, Mor Shpigel Nacson, Suriya Gunasekar, and Nathan Srebro. The implicit bias of gradient descent on separable data. The Journal of Machine Learning Research, 2018.
>
> [2] Mor Shpigel Nacson, Jason Lee, Suriya Gunasekar, Pedro Henrique Pamplona Savarese, Nathan Srebro, and Daniel Soudry. "Convergence of gradient descent on separable data." In The 22nd International Conference on Artificial Intelligence and Statistics. PMLR, 2019.

---

> > ### Comment · Reviewer_FypC · 2022-08-06
> > **Additional Questions**
> >
> > I want to thank the authors for the detailed responses. Overall, I am satisfied with the response and leans to accept this paper. However, I get some additional questions regarding the first response. The authors claim that "one of the goals of our paper is to demonstrate the practicality of $p$-GD for deep learning". However,
> >
> > 1. How is the (empirical) convergence speed of $p$-GD? An essential evaluation criterion for optimizers is the convergence speed, which is why Adam is still popular while practitioners have found it (usually) generalizes worse than SGD. It will be appreciated if the authors can provide the training loss curves of the experiments in the camera-ready version if accepted.
> >
> > 2. How is the compatibility of $p$-GD with other training techniques? I notice that in Line 707, the authors say "In particular, we find that not having weight decay costs us around $3%$ in validation accuracy in the $p = 2$ case". Does weight decay conflict with $p$-GD ($p\ne 2$)?
> >
> > All in all, I appreciate the theoretical efforts of this paper (I also find it very interesting (mathematically) if the result can be extended to MD with a larger class of potential functions). However, to attract the practitioners' interest in $p$-GD, the experiments in this paper are still a bit toy (only CIFAR 10 and Imagenet with low training accuracy). It may be a good idea to find a task and show that replacing GD with $p$-GD can achieve SOTA (this is definitely beyond the scope of this paper so please consider this as a suggestion for future work :)).
> >
> > A minor question: it seems that in the experiments, $p$-GD with $p=3$ outperforms GD consistently. Do the authors have any insight into this (just curious, it is totally fine if you do not know why)?

---

> > > ### Author Response · Authors · 2022-08-08
> > > **Thank you for your additional questions.**
> > >
> > > Thank you for these thoughtful comments. Let us answer your questions one by one:
> > >
> > > - We expect the empirical convergence rate of $p$-GD will be very comparable to that of gradient descent. For the final version, we will add the loss curves of $p$-GD with different values of $p$.
> > > - Training techniques such as weight decay are fully compatible with $p$-GD. We did not include weight decay to isolate the effects of different implicit biases induced by $p$-GD. We mentioned the lack of weight decay only to explain the difference between the performance of GD in our ImageNet experiment vs the standard baseline.
> > >
> > > Thank you for your suggestion for future experiments. We agree that it will be great to find a task and show that replacing GD with $p$-GD can beat the SOTA, and we will be on the lookout for that.
> > >
> > > Re minor question: We currently do not have an explanation on why $p$-GD with $p=3$ performs better, but it certainly merits further investigation in future work.

---

### Official Review · Reviewer_LBAU · 2022-07-09

**Rating:** 6
**Confidence:** 4
**Soundness:** 3 good
**Presentation:** 4 excellent
**Contribution:** 2 fair

**Summary:**

This paper studies the implicit bias of the solution found by mirror descent in the problem of classification with a linear model.
Similar problems have been studied for linear regression and/or gradient descent, but not simultaneously for mirror descent and classification.
To be fair, mirror descent is a very general optimization method, but here the paper considers a very specific setting, which they call "p-GD".
This stands for p-norm gradient descent, since (roughly speaking) the (p-1) power of the weights is used to update them.
Perhaps unsurprisingly, the main theoretical result is that p-GD finds the max-margin solution with respect to the p-norm, thus generalizing the previous finding that GD finds the max-margin wrt the 2-norm.
The results of some experiments are shown in support of the theoretical results.


**Questions:**

Minor:
- How do we know that the argmin in Eq.2 and the argmax in Eq.3 are unique?
- Eq.(p-GD), first row, w*sign(w) is just the absolute value of w, which implies that the value of the weight w_{t+1} is always positive.
Doesn't that limit drastically the solutions that can be possibly found by the optimizer?


**Limitations:**

OK

**Strengths And Weaknesses:**

Strenghts:
- The paper is very clear and well written.
- The study of implicit bias of solution found by optimization is an important problem.

Weaknesses:
- It is unclear to me how much this "p-GD" optimizer is relevant for the machine learning community, I don't know of anyone using it.
Section 4.2 reports an application of p-GD to deep neural networks and image classification but, as acknowledged by the authors, those results are "out of the scope of our theoretical results", so it remains a bit unclear what conclusions we are supposed to draw.

Overall my opinion is that the strengths slightly outweight the weaknesses, so I vote for a weak accept.

---

> ### Author Response · Authors · 2022-08-02
> **Author response to Reviewer LBAU**
>
> We thank the reviewer for their valuable comments and feedback. We would like to first clarify that one of the goals of our paper is to demonstrate the practicality of p-GD for deep learning. While mirror descent is a general family of optimization algorithms, as noted by the reviewer, it has not been adopted in deep learning (other than GD, of course). The primary reason for that is the MD update rule requires applying the inverse of the mirror map, which is computationally expensive, especially for deep neural networks with many parameters. However, the beauty of the special class of p-GD is that (1) the update rule becomes fully parallelizable and thus efficiently implementable in the same manner as GD; (2) different $p$ lead to weight vectors with significantly different geometries, and our experiments in Section 4.2 demonstrate that this leads to different generalization performance. Lastly, when we said “out of scope” in Section 4.2, we only meant that our theory does not directly predict the generalization performance of the trained classifier. We have rephrased this statement in the revision.
>
> To address the minor points:
> - The argmin and argmax in eq (2) and (3), respectively, are unique for $p > 1$. A proof of uniqueness in eq (3) can be found in Appendix B.5. A similar argument works for eq (2).
> - Thank you for spotting the typo in eq. (p-GD). The correct formula should be $|w_t^+[j]|^{\frac{1}{p-1}} \cdot \mathrm{sign}( w_t^+[j])$. So, the output of the update rule is not necessarily positive. We have corrected this in the revised version.

---

> > ### Comment · Reviewer_LBAU · 2022-08-07
> > **Thanks**
> >
> > Thank you for the response. I think that this work should be published in NeurIPS even if the relevance p-GD remains to be seen. It is likely that other researchers will continue studying the properties of p-GD in future work.

---

> > > ### Author Response · Authors · 2022-08-09
> > > **Thank you**
> > >
> > > Thank you again for your thoughtful comments. To further reflect some of the points that arose from this discussion, we made another round of updates to the paper (see all the changes in orange, such as line 71 in our contributions). We hope that these changes address the reviewer’s main comments.

---

### Author Response · Authors · 2022-08-02
**Thank you for yor comments and feedback**

We have uploaded a revision of our paper and highlighted our changes in orange. We hope that our response clarifies any remaining questions and concerns from the reviewers.

---

> ### Author Response · Authors · 2022-08-09
> **Thank you again!**
>
> Once again, we thank all the reviewers for the insightful discussion, which helped us further improve the paper. We have uploaded a new revision and highlighted all changes in orange.

---

### Meta-Review · Area_Chair_3tSu · 2022-08-29

**Recommendation:** Accept
**Confidence:** Certain

**Metareview:**

This paper studies mirror descent in the classification setting with exponential and logistic losses. The reviewers agreed that the problem is important, and the paper is clear and well written.

**Award:**

No

---

### Decision · Program_Chairs · 2022-09-14

Accept